# Nucleic Acid-Based Nanobiosensor (NAB) Used for *Salmonella* Detection in Foods: A Systematic Review

**DOI:** 10.3390/nano12050821

**Published:** 2022-02-28

**Authors:** Leticia Tessaro, Adriano Aquino, Paloma de Almeida Rodrigues, Nirav Joshi, Rafaela Gomes Ferrari, Carlos Adam Conte-Junior

**Affiliations:** 1Center for Food Analysis (NAL), Technological Development Support Laboratory (LADETEC), Federal University of Rio de Janeiro (UFRJ), Cidade Universitária, Rio de Janeiro 21941-909, RJ, Brazil; leticiatessaro@pos.iq.ufrj.br (L.T.); aquinolp@gmail.com (A.A.); paloma_almeida@id.uff.br (P.d.A.R.); nirav.joshi@ifsc.usp.br (N.J.); rafaelaferrari@yahoo.com.br (R.G.F.); 2Laboratory of Advanced Analysis in Biochemistry and Molecular Biology (LAABBM), Department of Biochemistry, Federal University of Rio de Janeiro (UFRJ), Cidade Universitária, Rio de Janeiro 21941-598, RJ, Brazil; 3Nanotechnology Network, Carlos Chagas Filho Research Support Foundation of the State of Rio de Janeiro (FAPERJ), Rio de Janeiro 20020-000, RJ, Brazil; 4Post-Graduation Program of Chemistry (PGQu), Institute of Chemistry (IQ), Federal University of Rio de Janeiro (UFRJ), Cidade Universitária, Rio de Janeiro 21941-909, RJ, Brazil; 5Post-Graduation Program of Veterinary Hygiene (PPGHV), Faculty of Veterinary Medicine, Fluminense Federal University (UFF), Vital Brazil Filho, Niterói 24230-340, RJ, Brazil; 6Physics Department, Federal University of ABC, Campus Santo André, Santo André 09210-580, SP, Brazil

**Keywords:** aptamer, biosensor, bacteria detection, nanomaterials, magnetic system

## Abstract

*Salmonella* bacteria is a foodborne pathogen found mainly in food products causing severe symptoms in the individual, such as diarrhea, fever, and abdominal cramps after consuming the infected food, which can be fatal in some severe cases. Rapid and selective methods to detect *Salmonella* bacteria can prevent outbreaks when ingesting contaminated food. Nanobiosensors are a highly sensitive, simple, faster, and lower cost method for the rapid detection of *Salmonella*, an alternative to conventional enzyme-linked immunosorbent assay (ELISA) and polymerase chain reaction (PCR) techniques. This study systematically searched and analyzed literature data related to nucleic acid-based nanobiosensors (NABs) with nanomaterials to detect *Salmonella* in food, retrieved from three databases, published between 2010 and 2021. We extracted data and critically analyzed the effect of nanomaterial functionalized with aptamer or DNA at the limit of detection (LOD). Among the nanomaterials, gold nanoparticles (AuNPs) were the most used nanomaterial in studies due to their unique optical properties of the metal, followed by magnetic nanoparticles (MNPs) of Fe_3_O_4_, copper nanoparticles (CuNPs), and also hybrid nanomaterials multiwalled carbon nanotubes (c-MWCNT/AuNP), QD/UCNP-MB (quantum dotes upconverting nanoparticle of magnetic beads), and cadmium telluride quantum dots (CdTe QDs@MNPs) showed excellent LOD values. The transducers used for detection also varied from electrochemical, fluorescent, surface-enhanced Raman spectroscopy (SERS), RAMAN spectroscopy, and mainly colorimetric due to the possibility of visualizing the detection result with the naked eye. Furthermore, we show the magnetic separation system capable of detecting the target amplification of the genetic material. Finally, we present perspectives, future research, and opportunities to use point-of-care (POC) diagnostic devices as a faster and lower cost approach for detecting *Salmonella* in food as they prove to be viable for resource-constrained environments such as field-based or economically limited conditions.

## 1. Introduction

*Salmonella* spp. is a bacterium of the Enterobacteriaceae family, a foodborne zoonotic pathogen that occurs mainly in foods of animal origin such as meat, eggs, seafood, and milk, when not subjected to adequate heat treatment. This can cause symptoms such as diarrhea, fever, and abdominal cramps after consumption and which in some severe cases can become fatal [1,2,3,4,5].

As for its taxonomy, there are two species of *Salmonella*, *S. bongori* and *S. enterica*. The latter comprises six subspecies, enterica, salamae, arizonae, diarizonae, houtenae, and indica. For each subspecies, several servares or serovars are found, which may exceed 2501 *Salmonella* serotypes, some being specific species (e.g., *Salmonella* Typhi specific to humans; *S.* Pollorum specific to birds) and others that can affect animals and humans (e.g., *Salmonella* Typhimurium), thus presenting a zoonotic character [6]. *Salmonella* Typhimurium is the main agent of human gastrointestinal infections [4,5,7,8]. These various serotypes have virulent genes. For example, *Salmonella* enterotoxin (Stn, 29 kDa protein (Stn)) is an agent for gastroenteritis that is responsible for the pathogenicity of *Salmonella* [4]. Dietary standards indicate that the tolerance for *Salmonella* in food is zero (absence in 10 or 25 g), due to the low infective dose (1UFC) and health risks, in addition to the high prevalence [9,10].

According to the Center for Disease Control and Prevention (CDC), when more than one person becomes ill from eating the same contaminated food or drink, the event is called an outbreak of foodborne disease [11]. These outbreaks are caused by pathogens, including germs (bacteria, viruses, and parasites), chemicals, and toxins. *Salmonella* is estimated to be responsible for about 1.35 million infections, 26,500 hospitalizations, and 420 deaths in the United States each year, with food being the main route of contamination [11]. 

Figure 1 presents the published studies for the detection of *Salmonella* in food during the years 2010 to 2020, by various techniques. The number of studies has been growing every year, evidencing the importance of monitoring the pathogen.

The techniques used to detect these genes are time-consuming and laborious [12]. Several detection methods were developed to avoid the consumption of contaminated foods with *Salmonella* [13]. The classic methods such as polymerase chain reaction (PCR) and enzyme-linked immunosorbent assays (ELISA) are used, but they have disadvantages such as time-consuming, use of large sample volume, expensive reagents, equipped laboratory, and labor-intensive, which limit their further application, making it necessary to develop the faster techniques that are low cost because the detection of foodborne pathogens at an early stage helps to prevent damage to human health [2,14,15]. Accordingly, biosensors are a simple platform to detect *Salmonella* spp. in foods [12]. The biosensor requires two steps: target recognition and signal detection [16].

Nanotechnology, in nanomaterials application, has received attention in the development of biosensors as analytical devices [9,14]. DNA detection methods using nanotechnology and biosensors (nanoparticle-based DNA biosensors) have been shown to be an excellent alternative for *Salmonella* detection in foods because of their simplicity [15,17]. Nanomaterials with different sizes and morphologies have been applied to biosensors [2]. The application of the nanomaterials for the detection occurs through the immobilization of biological recognition elements, for example, single-stranded DNA (ssDNA) or aptamer complementary to *Salmonella*, the target DNA on nanoparticle surface [18].

Biological recognition elements such as complementary DNA with 20–40 base pairs are used to hybridize several biosensors. DNA-based biosensors are highly selective to single-stranded DNA segments when immobilized in a way that retains stability, reactivity, accessibility to the target analyte, and optimal orientation on the electrode surface, producing an electrical signal when the DNA target (DNAt) binds to the complementary sequence of the capture (probe DNA) [18]. Biological recognition elements such as aptamers are short single-stranded (ss) nucleic acids of the DNA or RNA that can be used to bind from peptides to proteins to parasites, cells, or viruses. The nucleic acids can form three-dimensional folded structures to form binding pockets and clefts for the specific recognition and tight binding capable of specifically binding to targets. A typical aptamer is synthesized in vitro from vast combinatorial libraries with different sequences in a “systematic evolution of ligands by exponential enrichment” (SELEX) [16]. 

Aptamer-based nanosensors use the aptamer for biological recognition and have been developed to detect and quantify different compounds. This platform is used for fast detection methods and does not require any sample, making them simple and robust in addition to providing excellent sensitivity and selectivity [19]. The transducers convert detectable biological recognition signals such as color change, fluorescence signal, and electrochemical signals. Nanomaterials such as gold nanoparticle (AuNP) [20,21], magnetic nanoparticle (MNPs) [22], quantum dot (QDs), carbon nanotubes (CNTs) [23,24,25], and carbon nanotubes with coupled gold nanoparticles (MWCNTs/AuNPs) [26] are the signal transduction elements [16].

In this sense, this systematic review (SR) evaluated the use of biosensors with a nanomaterial aptamer or DNA to detect *Salmonella* spp. This systematic review aimed to evaluate the types of nanomaterials used and the sensitivity presented by biosensors to detect *Salmonella*, highlighting the use of the magnetic separation system. For this purpose, three scientific databases (Science Direct, SCOPUS, and Pubmed) were used for the selection of studies; data collection followed by evaluation of the influence of immobilization of biological recognition elements in the LOD (limit of detection) of the biosensor devices.

## 2. Methodology

This systematic review analyzed available three scientific databases (Science Direct, SCOPUS, and Pubmed) about DNA and aptamer-based biosensors to *Salmonella* spp. detection following a four-phase flow diagram and guidelines for systematic review and meta-analyses (PRISMA) [27]. 

### 2.1. Research Strategy and Data Extraction 

The selections of the articles were made through the individual reading of the titles and abstracts. The articles were excluded if the association between nanomaterial, biosensor, and *Salmonella* spp. is not investigated. Searches were performed on 5 January 2022, and limited to English-language, with a date delimited between 2010 and 2022. Articles, editorials, reviews, letters, and doctoral theses were excluded outside the stipulated period. In addition to the selected articles, the inclusion of articles considered essential to create the review was performed according to the established criteria.

The data were extracted from the selected articles, including data on the type of the nanomaterial, biorecognition element, transducer, *Salmonella* serotype, sensitivity, specificity, limit of detection, and type of samples.

### 2.2. Focus Questions

The focus issue was according to the population, intervention, comparison, and outcome (PICO) method. The research questions were based on the following form: (P)—What types of nanomaterials can be used to improve the detection of *Salmonella* spp.? (I)—Which nanoparticle is most used for the detection of Salmonella spp. using a biosensor? (C)—Which is the best nanomaterial for biosensors? (O)—What are the functions of nanomaterials in detecting bacteria in food?

### 2.3. Information Sources

A literature search was performed using Medical Subject Headings (MeSH) terms for Science Direct, SCOPUS, and Pubmed. The initial screening process was performed for the period 2010 to 2022. Further directed searching was also carried out by checking the reference list of relevant articles.

Search component 1 (SC1) included the following population search: (SC1a) Nanoparticle* OR Nanocrystalline Materials OR Nanocrystalline Material OR Nanocrystals* OR Nanocrystal* OR Nanostructure* OR “Nanostructured Material” OR Nanomaterial*; (SC1b) *Salmonella**. 

Search component 2 (SC2) included the following intervention search: “Biosensing Technique” OR Technique, Biosensing OR Techniques, Biosensing OR “Biosensing Technics” OR Biosensing Technic OR Biosensors* OR Biosensor* OR Electrodes, Enzyme OR Electrode, Enzyme OR “Enzyme Electrode” OR “Enzyme Electrodes” OR Bioprobes* OR Bioprobe* OR Biosensors* OR Biotechnology OR Electrodes* OR Monitoring, Physiologic/instrumentation.

## 3. First Visual Approaches to the Dataset

The evaluation of the articles was carried out independently by three reviewers. If there was doubt about the eligibility of the study, the authors did not exclude it and made the decision only after reading the complete text, analyzing whether they met the eligibility criteria or not. The results of the systematic search are presented in the PRISMA flowchart in the Figure 2.

Twenty-four articles were selected and read thoroughly. The differences observed are presented in Table 1 and used for the development of this review. The extracted data were: aptamer biorecognition material (n = 13) or DNA (n = 11), type of nanomaterial used, and LOD. The results of this systematic review are presented with a focus on evaluating the influence of nanomaterials and the type of biological recognition element on the LOD.

### 3.1. Salmonella *spp.* in Food

*Salmonella* spp. is a Gram-negative bacterium that belongs to the Enterobacteriaceae family and is one of the main foodborne pathogens. It is responsible for cases of gastroenteritis that are usually self-limiting; however, in some cases, the course of the disease can be more complicated, depending on the serotype, the response of the affected organism, infective dose, treatment used, when necessary, and aspects of the agent itself as virulence factors, invasiveness, and resistance to antimicrobials [43,44]. According to the CDC, this agent is implicated as the cause of approximately 1.35 million infections with 26,500 hospitalizations and 420 deaths per year in the United States [9,45].

Some of *Salmonella* serovars are specific to human species, such as *S*. Typhi and *S. Paratyphi* A, B, and C, and others are specific to animal species, such as *S. Dublin* for cattle, *S*. Choleraesuis, and *S*. Typhisuis for pigs, *S*. Abortusequi for horses, and *S. Pullorum* and *S*. Gallinarum for birds. However, other serovars have a zoonotic character, with Typhimurium and Enteritidis being the two most identified in the world, attributed to human contamination via food [46].

Among the main sources of contaminants are products of animal origin, such as beef, pork, chicken, milk, seafood, and eggs. *S*. Typhimurium is the most common serovar in meat products, and *S. Enteritidis* is found in eggs [13]. These bacteria can colonize the animal’s intestinal tract, with some being asymptomatic carriers and eliminated in the feces or having vertical transmission to the egg. Feces can directly contaminate other animals, vegetables, reach the animal carcass during improper handling of the intestinal tract in the industry, or be transmitted through the handling of food by asymptomatic carriers [47,48]. The survival of the pathogen in the food depends on the storage temperature and the consumption of raw or undercooked food [49,50].

Detection of *Salmonella* spp. can be done through traditional techniques such as plate cultivation and biochemical and serological tests to identify species, subspecies, and serovars. However, molecular methodologies tend to be techniques of choice for faster detection when compared to traditional microbiology techniques, in addition to their greater sensitivity and specificity [51]. However, in terms of detecting pathogens in the field, biosensors stand out as important tools for quick and practical identification.

### 3.2. Nanomaterial–Based Biosensor Used for Salmonella spp. Detection 

Nanomaterials are used in biosensors because of their valuable properties such as size, high surface to volume ratio [16], suitable biocompatibility, and surface effect [36], which help improve sensitivity. 

Most commonly used nanoparticles (NPs) in detection biosensors are those of noble metals such as gold, silver, and copper due to their optical properties and intense absorption plasmonic resonance surface (SPR) in the visible region, which have further improved their sensitivity and provide a new means of detection through direct visualization [16]. The shape of these can be of great diversity: spherical, nanorods, nanostars, and nanoflowers, among others. Gold nanoparticles (AuNPs) are widely used in the construction of biosensors because of their unique optical properties and the ability to signal amplify SPR, together with their biocompatibility and inertness [52]. Silver nanoparticles are also often used in the construction of biosensors because they have advantages such as their recognized antibacterial property, high surface area, and electron transport efficiency, besides being a more economical alternative compared to other metals [53]. 

The use of carbon-based nanomaterials is frequent in biosensors because of their properties such as mechanical resistance, high surface area, and excellent electrical conductivity, making them very sensitive when exposed to biomolecules. We can highlight graphene and carbon nanotubes and their modification with nanomaterials due to their wide application. The graphene oxide (GO) surface can be modified, providing a platform for the aggregation and binding of the AuNP [5]. Fe_3_O_4_/Au core–shell and AuNPs based biosensors, for example, have been built to detect several compounds in complex samples [36]. In another example, AuNP coupled to GO shows a color change of the solution after functionalization. AuNPs have surface plasmon resonance properties responsible for changing the color of the biosensor and which are affected by the distance between the nanoparticles and by DNA hybridization. [36]. 

Quantum dots (QDs) are among the emerging nanomaterials known for their ability to develop multitasking, such as medical diagnostics, drug administration, and genetherapy [54]. Structurally, QDs consist of a semiconductor nucleus coated by a shell, and the cover agents or binders are the determining factors in applications. Their use involves a variety of bioapplications, such as protein detection, DNA and other biomolecules, cell marking, and binding assays to investigate target events using fluorescence resonant energy transfer (FRET), along with use in biosensors for the detection of influenza virus [55] and pesticides [56]. 

Figure 3 presents some different nanomaterials that have been used in nanobiosensors for pathogen detection. The nanomaterials used for detection of *Salmonella* spp. in several foods [3,39] is shown in Table 1.

Different conducting polymers of various morphologies can be used in biosensor fabrication because they have several electrical properties; for example, polypyrrole (PPy) is a promising conducting polymer synthesized from a pyrrole monomer. Another example is PPy- functionalized reduced graphene oxide (PPy-rGO) nanomaterial that was prepared by chemical polymerization [3].

Quantum dots (QDs) are semiconductor nanocrystals that absorb high-energy photons with high quantum. The QDs have been applied as opposed to the traditional organic fluorophores. Another kind of nanomaterial is upconverting nanoparticles (UCNP) that absorb two or more photons and exhibit anti-Stokes type emission with strong, sharp, and visible luminescence [35].

Nanomaterials can be used in combination to improve the detection system. For example, gold nanoparticles (AuNPs) as colorimetric probes and magnetic nanoparticles (MNPs) are used to concentrate elements. AuNPs possess good biocompatibility and optical performance [42,57,58,59,60]. MNPs are popular separation materials and have been extensively implemented in the field of DNA hybridization analysis [42,61].

### 3.3. DNA-Based Nanosensor for Salmonella spp. Detection

DNA probe diagnostic testing has become a high-potential technology for pathogen detection analysis in food samples in recent years. Nanosensors with probes containing nucleic acids allow direct detection in the genetic fragment, fully recommended to be foodborne since there is no possibility of finding antigens or antibodies, as is usually done in physiological samples. The use of DNA coupled to nanomaterials in probes improves sensitivity and specificity and can often detect without amplifying bacterial RNA [18]. 

Most of the work selected in this SR uses only AuNPs immobilizing DNA to form the detection probe. Gold is the most used material for this function because of its biocompatibility and its SPR band appearing in the visible region, which often allows detection with the naked eye ([52,58]). AuNPs were used to improve the specificity of the probe to immobilize two different fragments in the nanoparticle for colorimetric detection of *S*. Choleraesuis, *S*. Typhimurium, and *Salmonella* specific with very high sensitivity, exhibiting a detection range from 6.76 to 50 nM with LOD 1 CFU mL^−1^ [15]. The species was tested using mismatched targets, called by the author M1, M2, and M3, and were designed with one, two, and three adjacent incompatibilities, presenting to be 100% specific to the bacterium of interest when tested at levels of 0–100 nM. Gold nanoparticles are functionalized with streptavidin (SA), forming the AuNPs-SA for the construction of a lateral flow nanosensor (LFA) [30]. Two sequences were used in this assay for greater specificity, and the detection limit for *Salmonella* detection was 3 × 10^3^ CFU mL^−1^ [30]. 

Saini et al. [4] functionalized strands of DNA to immobilized carbon walls (c-MWCNT) coupled to AuNPs. The results of *S*. Enterica obtained with conventional methods of detection in milk were compared with the nanosensor. The nanosensor with electrochemical detection was performed by cyclic voltammetry (CV) and differential pulse voltammetry (DPV) using GCE. The LOD obtained by the 0.3 pg-1 sensor was compared to other analysis methods, which showed to be better from selected studies in the literature. The sensor’s specificity was confirmed in the presence of other food pathogens: Klebsiella pneumoniae origin, *E. coli*, and *Lis-monocytogenes* of S Enterica in the presence of other pathogens applied in a sample of other pathogens [4].

The study by Vetrone et al. [18] developed a nanosensor to detect *S. Enteritidis*, belonging to the class of *S. enterica*, in milk and orange juice samples. The sensor takes the form of hybridized sandwiched samples (AuNPs-DNAt-MNPs) by the interaction between DNA-functionalized AuNPs and sample-specific DNA-DNA-functionalized MNPs, separating positive detections with a magnet without the need for prior PCR amplification. The transducer used was electrochemical and analyzed by detecting gold voltammetric peaks using differential pulse voltammetry. The LOD for milk and orange juice samples ranged from 1 to 100 ng mL^−1^, demonstrating excellent sensitivity; PCR did not perform because most PCR-amplified DNA detection methods are validated in the range up to 500 ng mL^−1^ [18]. 

The use of graphene oxide (GO) modifying glassy carbon electrode (GCE) was explored by Xu and collaborators in 2019. The hybridized DNA as nanoparticles of Fe_3_O_4_ was denoted as dsDNA/Fe_3_O_4_ NPs/CGO/GCE by using methylene blue (MB) as a redox indicator under optimal conditions to improve sensitivity and selectivity in the electrochemical detection of Typhoidal *Salmonella*. The authors compared their LOD obtained, 3.16 × 10^−18^ M, to other studies in the literature with modified GCE electrodes and obtained much better sensitivity when compared to these studies. The biosensor was validated using real serum samples that obtained SRD between 3.8% and 5.6%; recoveries ranged from 92.17% to 100.65%, which are promising for application in real samples [31].

A vitreous carbon electrode (GCE) modified with reduced polypyrrole graphene oxide nanocomposite (PPy-rGO) was developed by Ye et al. [3]. For signal amplification, biofunctionalized gold nanoparticles of horseradish peroxidase (AuNPs-HRP-SA) were used for signal amplification. The AuNPs were electrodeposited on PPy-rGO/GCE, forming the electrode AuNPs/PPy-rGO/GCE; the nanoparticles were functionalized with a complementary DNA sequence (cDNA) to give specificity to the genosensor. 

Evaluating the analytical performance of the nanobiosensors that were selected in this SR using c-DNA in functionalization presented suitable sensitivity in different samples such as milk, drinking water, and pork. Some studies stand out for their excellent sensitivity of 6.76 aM [32] and 0.3 pg mL^−1^ [4] by using AuNPS, c-MWCNT/AuNPs, and AuNPs applied to milk samples. Most of the studies also evaluated the specificity of the sensor to other pathogens, such as Klebsiella pneumoniae, E. coli, and Listeria monocytogenes, which is specific to target *Salmonella*. It is noted that AuNPS is the nanomaterial used the most. Additionally, studies with better sensitivity use it due to the unique properties of the noble metal Au.

### 3.4. Aptamer-Based Nanosensor for Salmonella spp. Detection

Functional aptamers, DNA, or RNA have become a key molecular tool for diagnosis. Due to their high specificity, selectivity, and high reproducibility when used in nanosensors, they are much more advantageous than the most natural receptors such as antibodies or enzymes. These advantages are related to the nature of the aptamer that instead selectively binds to a specific sequence, which can be synthesized with high purity, making them highly specific for detecting pathogens, such as *Salmonella* bacteria, in nanosensors [19]. As in the previous section, most of the studies selected in this SR used nanomaterial AuNPs to improve the sensitivity in detection due to their unique optical properties that employ optical transducers such as SERS, LSPR, fluorescence, electrochemical, and colorimetric.

Studies using SERS employ different morphologies of spherical and spiny NPs. Duan et al. [17] used spherical films with two different aptamers and presented LOD 27 CFU/mL showing specificity in the presence of multiple pathogens. 

Similarly, Ma et al. [14] used spiny NPs with a diameter of the particle size (40 nm) and a detection limit of 4 CFU/mL, specific to *S.* Typhimurium when tested with five different bacteria [2]. Colorimetric detection was shown to be less sensitive, LOD 10^5^ CFU/mL, but high specificity in the presence of *E. coli* (ATCC 25922, CMCC44825, and CMCC44151), *S. Paratyphi* A, *S. Paratyphi* B, *S. flexneri*, *P. aeruginosa*, *L. monocytogenes*, *S. aureus*, *S.* Typhimurium, and *E. coli* O157:H7 [41].

The study that presented better sensitivity among those selected with LOD 1 CFU mL^−1^ used an electrochemical transducer with GCE electrode modified with OG and deposited AuNPs. The nanosensor showed high selectivity when tested with the bacteria *L. monocytogenes*, *B. subtilis*, *S. aureus*, *S. pyogenes*, *E. coli*, and *E. sakazakii,* applied to pig samples and analyzed for 35 min.

Using more than one nanomaterial is a strategy that can improve sensitivity and selectivity in detecting the target pathogen. Duan et al. [17] used AuNPs of 13 nm functionalized with aptamer 1 (AuNPs-apt1) and also magnetic nanoparticles of Fe_3_O_4_ functionalized with aptamer 2 (MNPs-apt2) that in the presence of target *S.* Typhimurium formed a sandwich structure of AuNPs-apt1-target-MNPs-apt2 resulting in colorimetric change. The nanosensor presented LOD 10 CFU mL^−1^ and specificity in the presence of *E. coli*, *L. monocytogenes*, *V. parahaemolyticus*, and *S. aureus* [42]. Zhang et al. [36] also used this combination of nanomaterials to detect *Salmonella* Typhimurium with SERS transduction as a detector. It presented excellent specificity in the presence of *E. coli*, *V. parahaemolyticus*, *B. cereus*, *S. dysenteriae*, *S. aureus,* and *S.* Typhimurium, and LOD sensitivity 15 CFU mL^−1^ and applied to pork samples, showing to be a suitable alternative for detection [36].

Ren et al. [40] reported on two nanomaterials: CdTe quantum dots–Aptamer1 and Fe_3_O_4_ magnetic nanoparticles, to detect *Salmonella* Typhimurium. The type of transducer used for detection was fluorescence and obtained the best sensitivity of this class of aptamers, LOD 1 CFU mL^−1^. The nanosensor proved specific for *Salmonella* Typhimurium and other foodborne pathogens, including *Staphylococcus aureus*, *Escherichia coli* O157:H7, *Listeria monocytogenes*, *Bacillus cereus*, *Salmonella Enteritidis*, and *Pseudomonas aeruginosa*, as negative controls, were added to the system. The biosensor was validated using natural drinking water, and milk samples presented recovery from 95% to 110% with suitable analytical precision, RSD < 10%, an excellent alternative in pathogen detection.

The analytical performance of biosensors was excellent, in terms of sensitivity and specificity, due to the functionalized aptamer in the nanomaterial, which was extremely selective to the target. The selected studies with the best sensitivities were 1 CFU mL^−1^ [40], 3 CFU mL^−1^ [14], and 4 CFU mL^−1^ [2] using nanomaterial MNPs@CdTeQDs, AuNPs, and AuNPs, applied to milk and drinking water, pork, and pork samples respectively. The nanomaterial used most was AuNPs that obtained suitable LOD values; however, the most sensitive work of this class (LOD 1 CFU mL^−1^) used a nanohybrid with MNPs@CdTeQDs magnetic particles, which helped in sensitivity. In most of the studies, specificity was also evaluated for other pathogens, such as *E. coli*, *V. parahemolyticus*, *B. cereus*, *S. dysenteriae*, *S. aureus*, and *S.* Typhimurium. The sensors were sensitive and selective, becoming an excellent alternative for *Salmonella* detection in different matrices.

### 3.5. Nanosensors with Different Transducers for Detection of Salmonella

The role of transducers in nanosensors is to transform information into an analytical signal for reading the electrical or digital signal. Different types of transducers have been used for pathogen detection, such as fluorescence [39], ultraviolet [42], surface-enhanced Raman scattering (SERS) [36], electrochemical [14], LSPR [37], etc. Among them, SERS has recently gained increasingly more attention. In this section, we discuss those that are used most, as reported in the selected articles of this SR: electrochemical, colorimetric, SERS, and fluorescence. 

#### 3.5.1. Electrochemical Transducers

Nanosensors with electrochemical transduction are widely used due to their advantages, such as low cost, ease of use, portability, and simplicity of construction. It does not depend on the volume of reaction, requires minimal sample volumes, no sample preparation, and has excellent sensitivity. After the specific link between the biorecognition material and target analyte, a measurable current can be generated, resulting in the process of amperometry or voltammetry. If there is an accumulation of measurable or potential load, the process then will be potentiometry, or if there is an alteration of the conductive properties of the medium between electrodes, the process then will be conductometry [62]. 

Electrochemical transducers were used most in the studies selected in this SR, and transducers showed better sensitivity to nanomaterials functionalized with DNA in the range 6.76 aM to 100 ng and 8.07 CFU mL^−1^ with varied nanomaterials: c-MWCNT/AuNP, MNPs-DNA-AuNPs, Fe_3_O_4_-NPs/CGO/GCE, and AuNPsAs mentioned, the electrochemical technique presents excellent sensitivity, and this class of transducer presented the two studies with the best sensitivity: 3.16 and 6.76 aM by Xu et al. [31] with Fe_3_O_4_-NPs/CGO/GCE and Zhu et al. [32] with AuNPs, although its limitation is that it requires instrumentation capable of measuring the energy values generated to visualize the detection result.

#### 3.5.2. Transducer Colorimetric

The colorimetric transducer for detection of biomolecules is the most used due to the great advantage of presenting a visual change in the coloration, making the detection possible with the naked eye, which requires little or no instrumentation, especially when used with AuNPs due to the SPR phenomenon of this nanomaterial [63,64]. Various factors, including size, shape, morphology, and interparticle distance, influence the characteristics of this transducer [65]. The disadvantage of this transducer lies in the limitation when working with small concentrations, in this case with *Salmonella* spp. bacteria, making it difficult to read color signals and resulting in low detection sensitivity [57].

The studies selected in this SR with colorimetric transducers showed better sensitivity to nanomaterials functionalized with DNA, in the range <10–3000 CFU mL^−1^ and 21.78–50 ng, all of which used AuNPs. On the other hand, those functionalized with aptamers using AuNPs or MNPs did not present as suitable a sensitivity, resulting in a value of 10^5^ CFU mL^−1^. The colorimetric technique, as mentioned, presents excellent sensitivity, with the advantage of visualizing the detection result possible with the naked eye without using optical instrumentation.

#### 3.5.3. Transducer SERS

Transducer SERS technique is based on the amplified Raman response due to the interaction between an analyte and the plasma surface of metals. This amplified response generates a signal capable of detecting molecules at low concentrations without the need to prepare the sample [66]. When a nanomaterial binds to a biorecognition element and a molecule as a Raman reporter, it generates a complex known as antigen–SERS labels. This gives the transducer a sensitive and selective detection [66]. The studies selected in this SR with SERS transducers showed better sensitivity to nanomaterials functionalized with aptamers in the range 4 to 27 CFU mL^−1^ using AuNPs or MNPs. Therefore, SERS is easy to perform without instruments but requires care during operation and a good outline.

#### 3.5.4. Transducer Fluorescence

Fluorescence has advantages such as sensitivity, selectivity, and reduced detection time over methods that use absorbance, depending on the surrounding environment. Its sensitivity can reach 100 times greater than absorbance techniques due to interactions between fluorophores and superficial plasmons in metallic nanostructures [52], and its high selectivity occurs due to the fluorescent compound that usually has more than one emission spectrum [67]. The studies with this type of transducer selected in this SR showed excellent sensitivity, with the most sensitive nanosensor belonging to this class of transducers with LOD 1 CFU mL^−1^.

The studies selected in this SR with fluorescence transducers showed greater sensitivity to nanomaterials functionalized with DNA of 25 CFU mL^−1^ to CuNPs [33] and functionalized with aptamers in the range 1–464 CFU mL^−1^ and AuNPS or MNPs (Fe_3_O_4_) and CdTe QDs by Ren et al. [40], Wang and Kang [39], and Srinivasan et al. [38]. The fluorescence technique, as aforementioned, provides excellent sensitivity, but its limitation is that it requires optical instrumentation to visualize the detection data. 

### 3.6. Magnetic Separation System with Sample Preparation

Some studies selected in this SR stand out for using a magnetic separation system composed of magnetic particles functionalized with aptamer or DNA. This magnetic separation system allows the probe to detect the target and separate it by a magnetic tool, usually a magnet, making it unnecessary to amplify the genetic material. This also separates the target in the case of *Salmonella* from the probable residues in the medium. Figure 4 presents a general scheme of the use of this system.

Ren et al. [40] uses the magnetic separation system with a combination of Fe_3_O_4_ magnetic particles functionalized by aptamer (Apt-MNPs) and ssDNA2 marked with QD (complementary tape of the aptamer) for detection and quantification of *Salmonella* Typhimurium by fluorescence. Figure 5 shows a schematic representation of a fluorescent test combining magnetic nanoparticles and carboxyl CdTe QDs to detect *S.* Typhimurium. The biotin aptamer from *S.* Typhimurium was first coupled to the surface of streptavidin-labeled MNPs, and the amino-modified ssDNA2 was then labeled with carboxyl-capped CdTe QDs. The aptamer-ssDNA2 duplex, which works as a sensor detection probe, was formed by incubating the as-prepared CdTe QDs with Apt-MNPs. Apt-MNPs are employed to capture target *S.* Typhimurium, and CdTe QD-ssDNA2 were used as a signaling probe. After magnetic removal of the Apt-MNPs, a fluorescent signal was increased CdTe QDs (λexc/in = 327/612 nm), allowing the construction of an analytical curve in the concentration range 10 to 10^10^ CFU mL^−1^, and presenting the lowest LOD of the class, 1 CFU mL^−1^. The entire detection process can be carried out in 2 h and applied to food samples [40]. 

Park et al. [12] uses MNPs (Fe_3_O_4_) functionalized with specific aptamers and a substrate in the presence of H_2_O_2_. *S.* Typhimurium with colorimetric transduction allows for simple and rapid detection to the naked eye. The specific aptamers in the MNPs interact with *Salmonella* spp., consequently increasing the peroxidase activity of the MNPs, making the detection highly specific.

In the procedure (Figure 6), the MNPs were treated with DNA aptamers that blocked the surface of the MNPs, thereby inhibiting enzyme function. It is clear that interaction between the colorimetric substrate 3,3′,5,5′-tetramethylbenzidine (TMB) and the surface of the MNPs is required to promote oxidation with H_2_O_2_. Additionally, DNA aptamers aggregated MNPs, significantly limiting the ability of the surfaces and reducing their colorimetric property. Finally, when *Salmonella* was added to the solution containing MNPs and DNA aptamers, the specific DNA aptamers were separated from the MNPs due to their strong interaction with *S.* Typhimurium, and MNPs regained their peroxidase activity by re-exposing the surface to reagents and producing the blue-colored products.

After the detection process, a magnet is used for magnetic separation of other residues that may be present in the middle of the solution. The LOD 7.5 × 10^5^ CFU mL^−1^ obtained by this nanosensor is not as low as the other studies compared; however, it should be noted that colorimetric detection has the advantage of not needing optical instruments to verify detection.

The study by Duan et al. [42] presents excellent sensitivity even using colorimetric detection; the strategy of this study was to use nanohybrids AuNPs@MNPs to improve LOD. AuNPs are used as colorimetric probes and the MNPs as concentration elements, and the aptamers were first immobilized on the surface of AuNPs and MNPs, respectively. The time of incubation of *S.* Typhimurium so that the aptamer on the surface of the nanoparticles could bind specifically to the target and form a *Salmonella*-aptamer-AuNPs sandwich structure was 45 min. With the use of the magnet, a magnetic field was generated, and the complexes formed were easily separated from the solution, resulting in a fading of the suspension of AuNPs visually with the naked eye and a decrease in the ultraviolet/visible signal (UV/Vis). The linear range applied 25 to 10^5^ CFU mL^−1^ and LOD 10 CFU mL^−1^, showing high target sensitivity and can be applied to actual samples.

Vetrone et al. [18] developed a sandwich MNPs/DNA/AuNPs to detect *S. Enteritidis* and electrochemical detection using differential pulse voltammetry. The sandwich scheme generated after the interaction with the target was then separated by the magnet. The dets study results indicate that genomic DNA not amplified was detected at a concentration as low as 100 ng mL^−1^ from bacteria, similar to the reported detection levels amplified by PCR [18].

Studies using this magnetic separation system have excellent sensitivity and the advantage of not amplifying the genetic material before being applied to the biosensor. This makes it stand out from the others selected in this SR; magnetic separation systems can be widely used for different pathogens and matrices in biosensors because they reduce the cost of using standard amplification methods such as PCR, which are expensive and require highly qualified personnel.

## 4. Conclusions and Outlooks

The nucleic acid-based nanosensors of this SR were applied to detect *Salmonella* serotypes, such as *Salmonella* Choleraesuis, *Salmonella* Typhimurium, *Salmonella enterica*, and *Salmonella* Enteritidis in real milk matrices, pork, chicken, drinking water, and orange juice. The skewness of the nanosensor developed to the target serotype is due to the high affinity of the sequence of nucleic acids, aptamers, or ssDNA, of the target *Salmonella*. The nanomaterials used in nanosensors with the perception of improving sensitivity were not much varied, predominantly the use of AuNPs due to their unique optical properties of the metal, followed by MNPs of Fe_3_O_4_, CuNPs, and hybrid nanomaterials: c-MWCNT/AuNP, QD/UCNP-MB, and CdTe QDs@MNPs. The transducers used for detection also varied from electrochemical, fluorescent, SERS, RAMAN, and mainly colorimetric due to the possibility of visualizing the result of detection with the naked eye, without the need for any optical instrument, making it simple and robust when compared to others. The sensitivity of the nanosensor is directly linked to the type of nanomaterial employed and the type of transducer. Studies using nanohybrids showed excellent sensitivity: LOD 1 CFU mL^−1^ [40], 0.3 pg mL^−1^ [4], and 28 CFU mL^−1^ [35]. AuNPs were the most used, followed by MNPs. The magnetic separation system, with the MNPs, stood out in this review since it is not necessary to amplify the genetic material before detection in the nanosensor. The use of this system also showed suitable sensitivity: LOD 1 CFU mL^−1^ [41], 10 CFU mL^−1^ [42], and 7.5 × 10^5^ CFU mL^−1^ [12]. 

It was observed that the nanosensors developed were mostly not applied to point-of-care devices (POCs) as an alternative proposal for future perspectives. POCs are an alternative for the rapid diagnosis of various infectious and noninfectious diseases, standing out for rapid detection [68,69,70,71]. These devices can provide clinical diagnostics and information about bacterial exposure, making them useful for an outbreak scenario. Nanosensors are the first step toward a viable method of detecting bacterial pathogens, particularly for environments with limited resources, such as field-based or economically limited conditions.

## Figures and Tables

**Figure 1 nanomaterials-12-00821-f001:**
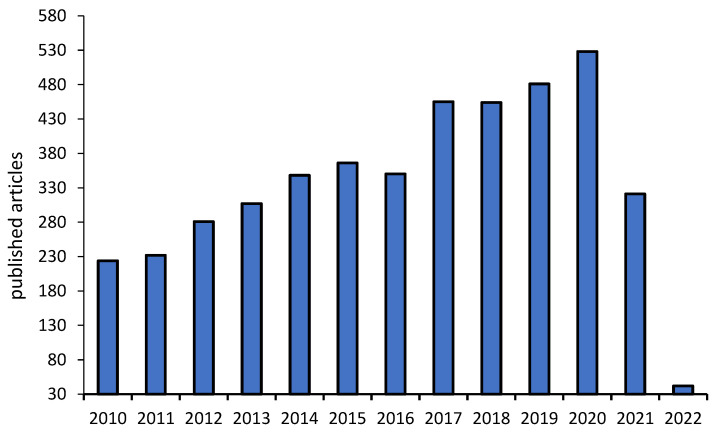
Published articles in the database Web of Science to detect *Salmonella* in foods in the years 2010–2022 as of February 3.

**Figure 2 nanomaterials-12-00821-f002:**
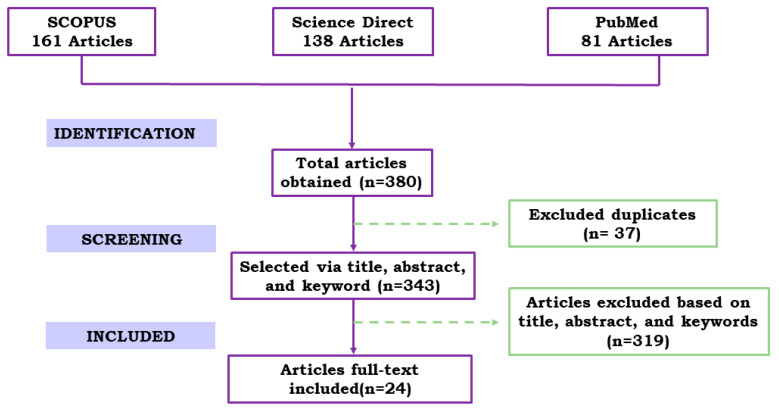
PRISMA flow chart showing the results of the systematic search for the period 2010 and 2022.

**Figure 3 nanomaterials-12-00821-f003:**
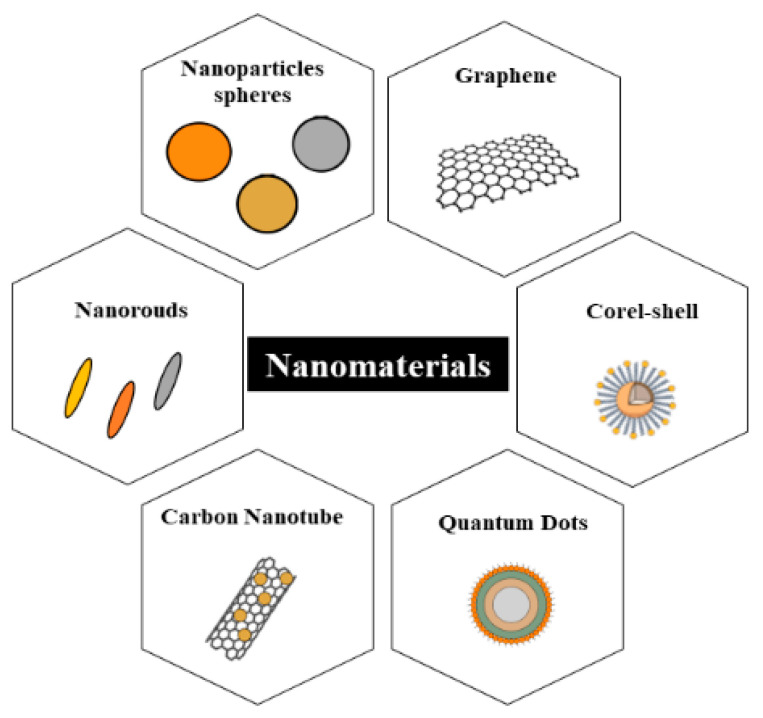
Different nanomaterials that have been used in nanobiosensors for pathogen detection.

**Figure 4 nanomaterials-12-00821-f004:**
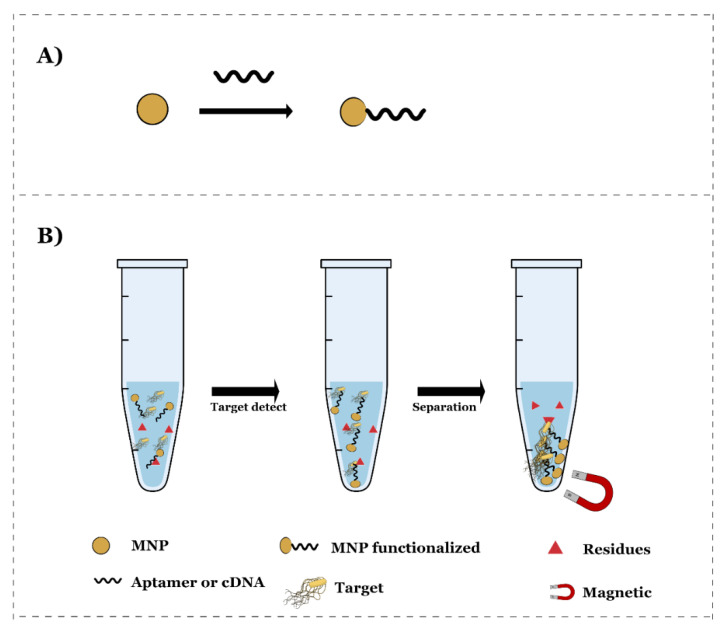
Schematic diagram of (**A**) functionalized magnetic nanoparticles (MNPs) with aptamer or complementary DNA (cDNA). (**B**) Illustration of the detection of *Salmonella* ssp. target with magnetic separation system.

**Figure 5 nanomaterials-12-00821-f005:**
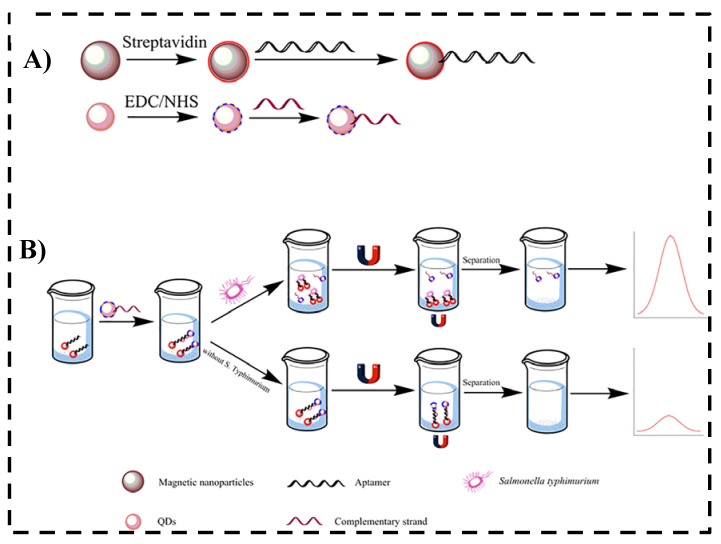
Schematic illustration of (**A**) synthesis of streptavidin magnetic nanoparticles and carboxyl CdTe QDs for detection *Salmonella.* (**B**) Detection of *S.* Typhimurium using prepared magnetic nanoparticles and CdTe QDs. Reprinted with permission from Ren et al. [40]. Copyright 2019, *PLoS ONE*.

**Figure 6 nanomaterials-12-00821-f006:**
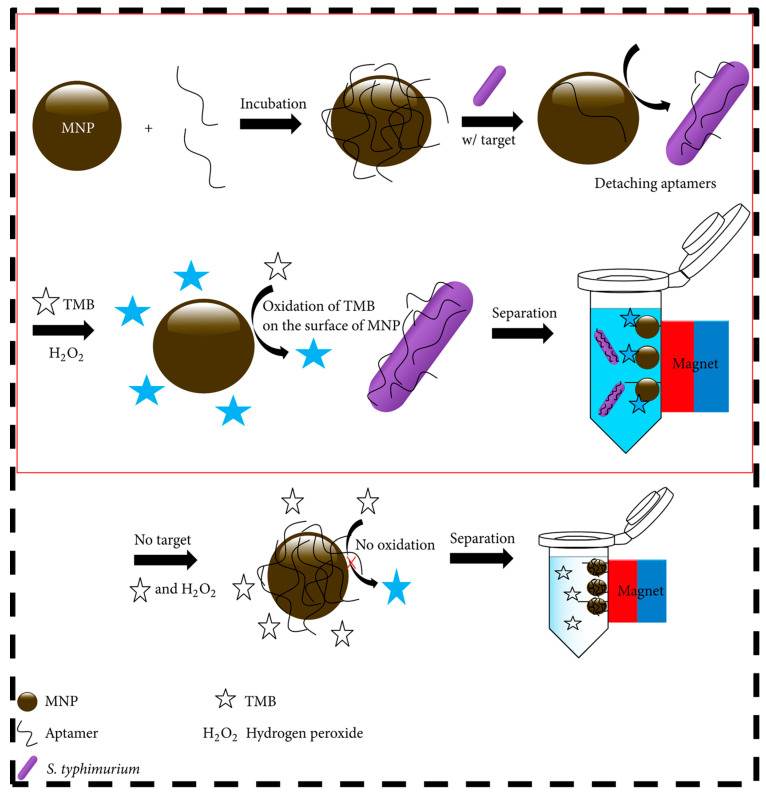
Schematic illustration of *Salmonella* Typhimurium detection using magnetic nanoparticles and colorimetric substrate (TMB) in the presence of H_2_O_2_. Reprinted with permission from Park et al. [12]. Copyright 2015, Hindawi.

**Table 1 nanomaterials-12-00821-t001:** Description of nanobiosensors.

Biorecognition Material	*Salmonella* Species	Nanomaterial	Type Transducer	Linear Range	Sequence	LOD	Sample	Ref.
**DNA**	*Salmonella* Choleraesuis	AuNPs-1 AuNPs-2	Colorimetric	-	5′-HS-AAAAAAAAAACTTAGCTGACATCATG-3′ (imm1) 5′-CGAGTCAGAGTAGTTTAAAAAAAAAA-SH 3′ (imm2)	50 nM	-	[15]
	*Salmonella* Typhimurium	AuNPs	Colorimetric	-	5′-GAACGGCGAAGCGTACTGGAA-3 (RP) 5′-CATCGCACCGTCAAAGGAACC-3′ (FP)	21.78 ng/mL	-	[28]
	*Salmonella* spp.	AuNPs	Colorimetric	10–10^3^ CFU/mL	5′ -ACCCACGCGTTTCATCGGTT-3′ 5′ -GCCGGCAATCCCTATCACCC-3′	<10 CFU/mL		[29]
	*Salmonella* Enteritidis	AuNPs-SA	Colorimetric	10^2^–10^7^ CFU/mL	5′CGGGGAGGAAGGTGTTGTGGTTAATAACCGCAGCAATTGACGTTA CC-3′	3 × 10^3^ CFU/mL	-	[30]
	*Salmonella* enterica	c-MWCNT/AuNP	Eletrochemical	0–31.7 pg/μL	5′-GTCCGGGTCAGCCTGAAT -3′	0.3 pg/mL	milk	[4]
	*Salmonella* enterica	MNPs-DNA-AuNPs	Eletrochemical	7–50 ng/mL	5′-CTAACAGGCGCATACGATCTGACA-3 (FP) 5′-TACGCATAGCGATCTCCTTCGTTG-3′ (RP)	<100 ng/mL	Milk and orange juice	[18]
	typhoidal *Salmonella*	Fe_3_O_4_-NPs/CGO/GCE	Eletrochemical	1–1 × 10^−8^ nmol/L	5′-GGCGGCGGGCGTCGCGCACG-3′	3.16 aM		[31]
	*Salmonella*	AuNPs-HRP-SA	Eletrochemical	9.6–9.6 × 10^4^ PFU/mL	5′-TCGGCATCAATACTCATC-3′	8.07 PFU/mL	-	[3]
	*Salmonella*-specific	AuNPs	Eletrochemical	10 aM–10 pM	5′-GCATCCGCATCAATAATACCG-3′ (FP) 5′ TTCTCTGGATGGTATGCCC-3′ (RP)	6.76 aM	milk	[32]
	*Salmonella* Enteritidis	CuNPs	Fluorescence	50–10^4^ CFU/mL	5′-TACCAAAATGTTGGATTGGATGTTGTACTGGGTTGCA-3′	25 CFU/mL		[33]
	*Salmonella* Typhimurium	AuNPs	Reflectivity	1 × 10^3^–1 × 10^8^ ng/mL	HS-T10-CAATCCGGACTACGACGCAC (CP) TTTATGAGGTCCGCTTGCTCTTTTTT-SH (DP)	0.01−100 ng/mL	-	[34]
**APTAMER**	*Salmonella*-specific	AuNPs	Eletrochemical	2.4–2.4 × 10^3^ CFU/mL	5′-HS-TATGGC GGC GTC ACC CGA CGG GGA CTT GAC ATT ATG ACA-G-3′.	3 CFU/mL	pork	[14]
	*Salmonella* Typhimurium	QD/UCNP-MB	Luminescence	50–10^6^ CFU/mL	5′-TATGGCGGCGTCACCCGACGG GGACTTGACATTATGACAG-3′ 5′-GGCGGTGTGAGGCTGGGAGGACGGACTGGG-3′ (cDNA)	28 CFU/mL		[35]
	*Salmonella* Typhimurium	AuNPs (Apt-Au-PDMS film)	SERS	27–2.7 × 10^5^ CFU/mL	5′-SH-AGTAATGCCCGGTAGTTATTCAAAGATGAGTAGGAAAAGA-3′	27 CFU/mL	-	[17]
	*Salmonella* Typhimurium	AuNPs	SERS	10^1^–10^5^ CFU/mL	5′ -TATGGCGGCGTCACCCGACGGGGACTTGACATTATGACA G-3′	4 CFU/mL.	pork	[2]
	*Salmonella*	MGNPs (Fe_3_O_4_) and AuNPs	SERS	10^2^–10^7^ CFU/mL	5′-SH-TAT GGC GGC GTC ACC CGA CGG GGA CTT GAC ATT ATG ACA G-3′	15 CFU/mL	pork	[36]
	*Salmonella* Typhimurium	AuNPs	LSPR	10^4^–10^6^ CFU/mL	5′-TATGGCGGCGTCACCCGACGGGGACTTGACATTATGACAG-SH-3′	10^4^ CFU/mL	pork	[37]
	*Salmonella* Typhimurium	AuNPs	Fluorescence	1.5 × 10^2^–9.6 × 10^4^ CFU/mL	5′-CCAAAGGCTACGCGTTAACGTGGTGTTGG−3′(Apt1) 5′-ATAGGAGTCACGACGACCAGAAAGTAATGCCCGGTAGTTATTCAAAGATGA GTAGGAAAAGATATGTGCGTCTACCTCTTGACTAAT-3′(Apt2)	464 CFU/mL		[38]
	*Salmonella* Typhimurium	SA-FSiNPs	Fluorescence	-	5′-biotin-(CH2)_6_-AGTAATGCCCGGTAGTTATTCAAAGATGAGTAGGAAAAGA-3′ 5′-biotin-(CH2)_6_-TGTCATGACCCGTAGGTAGTCTTAGAAGACTAGGCACGTT-3′	80 CFU/mL		[39]
	*Salmonella* Typhimurium	MNPs (Fe_3_O_4_) and CdTe QDs	Fluorescence	10–10^10^ CFU/mL	5′-biotin-C6-TATGGCGGCGTCACCCGACGGGGACTTGACATTATGACAG-3′(ssDNA1) 5′-C6-NH2-CTGTCATAATGTCAAGTC-3′(ssDNA2)	1 CFU/mL		[40]
	*Salmonella* Typhimurium	AuNPs	Colorimetric	-	50 -CCAAAGGCTACGCGTTAACGTGGTGTTGG −30	10^5^ CFU/mL	-	[41]
	*Salmonella* Typhimurium	MNPs (Fe_3_O_4_)	Colorimetric	-	5′-GAGGAAAGTCTA- TAGCAGAGGAGATGTGTGAACCGAGTAA-3	7.5 × 10^5^ CFU/mL		[12]
	*Salmonella* Typhimurium	AuNPs and MNPs (Fe_3_O_4_)	UV/Vis	25 to 10^5^ CFU/mL	5′-SH-ATAGGAGTCACGACGAC-CAGAAAGTAATGCCCGGTAGTTATTCAAAGATGAGTAG-GAAAAGATATGTGCGTCTACCTCTTGACTAAT-3′ (apt 1) 5′-Bio-ATAGGAGTCACGACGACCAGAAAGTAATGCG-CGGTAGTTATTCAAAGATGAGTAGGAAAAGATATGTGC-GTCTACCTCTTGACTAAT-3′ (apt2)	10 CFU/mL	milk	[42]

Legend: (-) not reported; AuNPs (gold nanoparticles); MB: magnetic beads UCNP: upconverting nanoparticle; QD: quantum dot; cDNA: DNA complementary; SGNPs (spiny gold nanoparticles); MGNPs (magnetic gold nanoparticles); MNPs (Fe_3_O_4_) (magnetic nanoparticles of Fe_3_O_4_); CdTe QDs (cadmium telluride quantum dots); UV/Vis (ultraviolet–visible); QD/UCNP-MB (quantum dots upconverting nanoparticle of magnetic beads); MNPs-DNA-AuNPs (DNA-coupled magnetic nanoparticles and sandwich-shaped gold nanoparticles); Fe_3_O_4_-NPs/CGO/GCE (graphene oxide modifying glassy carbon electrode coupled with oxide iron nanoparticles); Apt-Au-PDMS film (aptamer–Au nanoparticles–polydimethylsiloxane film); c-MWCNT/AuNP FP (carboxylated multiwalled carbon nanotube and gold nanoparticle); AuNPs-SA (gold nanoparticles–streptavidin); SA-FSiNPs (streptavidin-conjugated nanoparticle silica fluorescence); AuNPs-HRP-SA (horseradish peroxidase–streptavidin biofunctionalized gold nanoparticles); CuNPs (copper nanoparticles); CFU (colony-forming unit); PFU (plaque-forming unit); FP (forward primer); RP (reverse primer); CP (capture probe); DP (detection probe); *Salmonella* spp. (*S*. Agona, *S*. Anatum, *S*. Berta, *S*. Derby, *S*. Dublin, *S*. Enteriditis, *S*. Gallinarum, *S*. Heidelberg, *S*. Infantis, *S*. Javiana, *S*. Kentucky, *S*. Mbandaka, *S*. Montevideo, *S*. Muenster, *S*. Newport, *S*. Oranienburg, *S*. Saintpaul, *S*. Senftenberg, *S*. Thompson, and *S*. Typhimurium); Apt-MNPs (aptamer-coated Fe_3_O_4_ magnetic particles); LSPR (localized surface plasmon resonance); SERS (surface-enhanced Raman scattering).

## Data Availability

Not applicable.

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
