# Peer review of "Nucleic Acid-Based Nanobiosensor (NAB) Used for Salmonella Detection in Foods: A Systematic Review"

_nanomaterials, 2022, doi:10.3390/nano12050821_

Round 1

Reviewer 1 Report

Dear Editor,

The present letter is referring to the  manuscript nanomaterials-1605809 Review type, entitled  “Nucleic acid-based nanobiosensor (NAB) used for Salmonella detection in foods: A systematic review”. The authors searched an analyzed the literature related to nucleic acid-based nanobiosensors with nanomaterials to detect Salmonella from 2010 to 2021, they organized and present a well structured review of the topic, however there are some points that should be adressed before acepting the manuscript to be publish at Nanomaterials. The points are listed below:

Please verify the abreviations, not all the abbreviations have the meaning ( for example SERS, PCR, ELISA to mention some).

Please verify the citations in the text.

Page 11, paragraph 3 :” The use of graphene oxide (GO)….promising for application in real sample, there is not reference in the text

Page 12, paragraph 2 “ Studies using SERS….E. coli O157:H7” pleas verify the references

Session 3.5.1, the authors could indicate which of the sensors  present the best analytical parameters and include the respective references.

Session 3.5.4, paragraph 2 the authors should include the references for each of the mentioned data.

The Figure 3 should be Figure 4, please verify and correct

The reviewer recommends minor revision before accepting the manuscript for publication.

Sincerely,

The reviewer

Author Response

Reviewer #1: The present letter is referring to the manuscript nanomaterials-1605809 Review type, entitled “Nucleic acid-based nanobiosensor (NAB) used for Salmonella detection in foods: A systematic review”. The authors searched an analyzed the literature related to nucleic acid-based nanobiosensors with nanomaterials to detect Salmonella from 2010 to 2021, they organized and present a well structured review of the topic, however there are some points that should be adressed before acepting the manuscript to be publish at Nanomaterials. The points are listed below:

Please check the abbreviations, not all abbreviations have meaning (e.g. SERS, PCR, ELISA to name a few).

Answer: Thank toy for the observation. All abbreviations have been checked and corrected.

Page 11, paragraph 3:” The use of graphene oxide (GO)….promising for application in real sample, there is not reference in the text

Answer: Thank you for the observation. The reference was included and have marked in red. Page 14.

Page 12, paragraph 2 “ Studies using SERS….E. coli O157:H7” pleas verify the references

Answer: Thank you. The reference was verified.

Session 3.5.1, the authors could indicate which of the sensors present the best analytical parameters and include the respective references.

Answer: Thank you for the observation. The analytical parameters were included and have marked in red. Page 18.

Session 3.5.4, paragraph 2 the authors should include the references for each of the mentioned data.

Answer: Thank you. The reference was included. Page 19.

The Figure 3 should be Figure 4, please verify and correct

Answer: Thank you for the observation. The order of the figures has been corrected.

The reviewer recommends minor revision before accepting the manuscript for publication.

Answer: Thank you. We hope that our manuscript is now suitable for this reputed journal.

Reviewer 2 Report

This paper (Nanomaterials) “Nucleic acid-based nanobiosensor (NAB) used for Salmonella detection in foods: A systematic review” is interesting. However, before publication there are some important aspects that must be considered. Thus, it is my opinion that the manuscript needs a Major Revision before publication.

Comments:

An English revision is necessary – see attached file for suggestions.

  1. Acronyms must be defined when first used. For example, in the abstract “MNPs”, “c-MWCNT/AuNP”, “QD/UCNP-MB” and “CdTe QDs@MNPs”

  1. In the abstract “Magnetic Separation System” changes to “magnetic separation system”

  1. In the page 3, in the second paragraph “damageto” changes to “damage to”.

  1. In the section methodology “four scientific databases (ScienceDirect, SCOPUS, and Pubmed)” changes to “three scientific databases (ScienceDirect, SCOPUS, and Pubmed)”.

  1. In the section 3, which table was used?

  1. In the section 3.2, in the second paragraph “plasmonic resonance surface SPR” changes to “surface plasmon resonance (SPR)”.

  1. In the section 3.2, in the second paragraph “visee region” changes to “in the visible region”

  1. In the section 3.2, in the third paragraph, the sentence "AuNPs have surface plasmon resonance properties responsible for changing the color of the biosensor and that are affected by the distance between nanoparticles and DNA hybridization." needs a revision and a reference

  1. The Table 1 caption needs to be rewritten.

  1. The Table 1 needs to be improved.

  1. In the section 3.3, the third paragraph needs to be rewritten.

  1. In the section 3.3, vitreous carbon electrode changes to GCE.

  • According to the IUPAC, all physical quantities should be written in italics (and their corresponding subscripts and superscripts not in italics).

  • Please, check for some imperfections in citation style in the list of references.

  • Both some text imperfections (typos) and English language at some places should be corrected during a careful proof-reading.

Author Response

Reviewer 2#:

This paper (Nanomaterials) “Nucleic acid-based nanobiosensor (NAB) used for Salmonella detection in foods: A systematic review” is interesting. However, before publication there are some important aspects that must be considered. Thus, it is my opinion that the manuscript needs a Major Revision before publication.

Answer: Dear reviewer, we are happy that you liked the theme proposed in our work. We appreciate your availability to evaluate and for the significant contributions raised. Below, we responded to each of your comments.

 Comments:

 An English revision is necessary – see attached file for suggestions.

 Answer: Thank you fot the observation. A native speaker reviewed the manuscript

  1. Acronyms must be defined when first used. For example, in the abstract “MNPs”, “c-MWCNT/AuNP”, “QD/UCNP-MB” and “CdTe QDs@MNPs”

Answer: Thank toy for the observation. All abbreviations have been checked and corrected.

  1. In the abstract “Magnetic Separation System” changes to “magnetic separation system”

 Answer: Thank you. The observation was modified.

  1. In the page 3, in the second paragraph “damageto” changes to “damage to”.

Answer: Thank you. The sentence was modified.

  1. In the section methodology “four scientific databases (ScienceDirect, SCOPUS, and Pubmed)” changes to “three scientific databases (ScienceDirect, SCOPUS, and Pubmed)”.

  Answer: Thank you. The observation has been corrected

  1. In the section 3, which table was used?

 Answer: All discussion of session 3 was used table 1.

  1. In the section 3.2, in the second paragraph “plasmonic resonance surface SPR” changes to “surface plasmon resonance (SPR)”.

 Answer: Thank you. The sentence has been corrected

  1. In the section 3.2, in the second paragraph “visee region” changes to “in the visible region”

 Answer: Thank you. The sentence has been corrected

  1. In the section 3.2, in the third paragraph, the sentence "AuNPs have surface plasmon resonance properties responsible for changing the color of the biosensor and that are affected by the distance between nanoparticles and DNA hybridization." needs a revision and a reference

 Answer: Thank you for the observation.  The paragraph has been rewritten and referenced.Page 7

  1. The Table 1 caption needs to be rewritten.

 Answer: Thank you. The table has been improved.

  1. The Table 1 needs to be improved.

  Answer: Thank you. The table has been improved.

  1. In the section 3.3, the third paragraph needs to be rewritten.

  Answer: Thank you. The paragraph has been rewritten. Page 13.

  1. In the section 3.3, vitreous carbon electrode changes to GCE.

 Answer: Thank you. The sentence has been corrected

 According to the IUPAC, all physical quantities should be written in italics (and their corresponding subscripts and superscripts not in italics).

  Answer: Thank you for the observation. All physical quantities has been corrected.

  • Please, check for some imperfections in citation style in the list of references.

  Answer: Thank you for the observation. The list of references were verified

Reviewer 3 Report

Tessaro et al. have made a systematic review on nucleic acid-based nanobiosensor for Salmonella detection in foods. It is a timely and important review article which has been fairly justified in the presentation. However, there are several issues, as shown below, should be addressed to improve the quality of this review article for publication in Nanomaterials:

  1. Figure 2 – while 37 articles are subtracted as duplicates from 380 totally identified articles, what is the reason for excluding 319 articles to narrow down finally to only 24 articles? This should be specified.
  2. Table 1 – the first column details are not completely visible and the tabulation area should be adjusted to bring all the columns within in the printable margins.
  3. Table 1 – modify “Type” with “Salmonella species”.
  4. Table 1 – all the abbreviations used in this table should be explained in full form in the footnote, especially, CFU and PFU as well as all the nanoparticle and transducer abbreviations. CFU and PFU should be either in capital or small letter adopting one way throughout.
  5. Section 3.5 – Why “SERS” discussed under colorimetric and fluorescence transducers? Is not the full form of “SERS” is “Surface enhanced Raman scattering”.
  6. Section 3.5.2 & 3.5.3 – “SERS does not require optical instrumentation” in section 3.5.2 and “SERS requiring high-cost instrumentation”. It appears there is some with understanding of SERS by the authors.
  7. Section 3.6 – Consider modifying the sub-heading “Highlights”.
  8. Figures – in addition to three schematic figures drawn, the authors should identify some more figures (2 or 3) from the key reported findings and include them in this manuscript. The permission to reproduce can be easily obtained from the respective publishers (for example through RIGHTS LINK for Elsevier).
  9. Conclusions – this section should be presented as two paragraphs with one paragraph summarizing key findings/observations based on those 24 articles discussed, while the other paragraph highlighting the research gap in the form of future perspective.
  10. All the scientific names mentioned throughout the text should be italicized.
  11. The manuscript should be checked for typographical errors as well as values and units.
  12. There are more than one font types and font sizes were used in the manuscript which should be carefully checked and formatted appropriately.

Author Response

Reviewer 3#: Salmonella detection in foods. It is a timely and important review article which has been fairly justified in the presentation. However, there are several issues, as shown below, should be addressed to improve the quality of this review article for publication in Nanomaterials:

Answer: Dear reviewer, we are happy that you liked the theme proposed in our work. We appreciate your availability to evaluate and for the significant contributions raised. Below, we responded to each of your comments.

  1. Figure 2 – while 37 articles are subtracted as duplicates from 380 totally identified articles, what is the reason for excluding 319 articles to narrow down finally to only 24 articles? This should be specified.

Answer: Thank you for the observation. Reasons for the exclusion were included to figure 2. Page 5.

  1. Table 1 – the first column details are not completely visible and the tabulation area should be adjusted to bring all the columns within in the printable margins.

Answer: Thank you. The table has been arranged to present a complete view of the data.

  1. Table 1 – modify “Type” with “Salmonella species”.

Answer: Thank you. The observation was modified.

  1. Table 1 – all the abbreviations used in this table should be explained in full form in the footnote, especially, CFU and PFU as well as all the nanoparticle and transducer abbreviations. CFU and PFU should be either in capital or small letter adopting one way throughout.

Answer: Thank you. All the abbreviations were corrected in the manuscript.

  1. Section 3.5 – Why “SERS” discussed under colorimetric and fluorescence transducers? Is not the full form of “SERS” is “Surface enhanced Raman scattering”.

Answer: Thank you for the observation. The affirmations have been corrected and have marked in red.

  1. Section 3.5.2 & 3.5.3 – “SERS does not require optical instrumentation” in section 3.5.2 and “SERS requiring high-cost instrumentation”. It appears there is some with understanding of SERS by the authors.

Answer: Thank you for the observation. Sorry for the mistake, the affirmation was corrected. Page 18.

  1. Section 3.6 – Consider modifying the sub-heading “Highlights”.

Answer: Thank you. The session was modified for Magnetic separation system with sample preparation. Page 19.

  1. Figures – in addition to three schematic figures drawn, the authors should identify some more figures (2 or 3) from the key reported findings and include them in this manuscript. The permission to reproduce can be easily obtained from the respective publishers (for example through RIGHTS LINK for Elsevier).

Answer: Thank you for your suggestions, which we implemented in the revised version, especially with new figures (Figure 5 and 6). Pages 19 and 20.

  1. Conclusions – this section should be presented as two paragraphs with one paragraph summarizing key findings/observations based on those 24 articles discussed, while the other paragraph highlighting the research gap in the form of future perspective.

Answer: Thank you for the suggestion. We have revised the manuscript accordingly. Page 22.

  1. All the scientific names mentioned throughout the text should be italicized.

Answer: Thank you for the observation. All manuscript was revised.

  1. The manuscript should be checked for typographical errors as well as values and units.

Answer: Thank you for the observation. All manuscript was revised and have marked in red.

  1. There are more than one font types and font sizes were used in the manuscript which should be carefully checked and formatted appropriately.

Answer: Thank you for the observation. All manuscript was revised for font and size.

Round 2

Reviewer 3 Report

The authors have carefully addressed all the comments raised by reviewers and therefore I recommend acceptance of this article for publication in Nanomaterials.